# Hepatic stellate cells limit hepatocellular carcinoma progression through the orphan receptor endosialin

Carolin Mogler[1,2,3,†], Courtney König[1,4,†], Matthias Wieland[1,4], Anja Runge[1,4], Eva Besemfelder[1], Dorde Komljenovic[5], Thomas Longerich[6], Peter Schirmacher[2] & Hellmut G Augustin[1,4,7,*]

## Abstract

**Hepatocellular carcinoma (HCC) is among the most common and deadliest cancers worldwide. A major contributor to HCC progression is the cross talk between tumor cells and the surrounding stroma including activated hepatic stellate cells (HSC). Activation of HSC during liver damage leads to upregulation of the orphan receptor endosialin (CD248), which contributes to regulating the balance of liver regeneration and fibrosis. Based on the established role of endosialin in regulating HSC/hepatocyte cross talk, we hypothesized that HSC-expressed endosialin might similarly affect cell proliferation during hepatocarcinogenesis. Indeed, the histological analysis of human HCC samples revealed an inverse correlation between tumor cell proliferation and stromal endosialin expression. Correspondingly, global genetic inactivation of endosialin resulted in accelerated tumor growth in an inducible mouse HCC model. A candidate-based screen of tumor lysates and differential protein arrays of cultured HSC identified several established hepatotropic cytokines, including IGF2, RBP4, DKK1, and CCL5 as being negatively regulated by endosialin. Taken together, the experiments identify endosialin-expressing HSC as a negative regulator of HCC progression.**

**Keywords** cancer; HCC; stromal cross talk; tumor stroma; vascular biology
**Subject Categories** Cancer; Digestive System

## Introduction

Hepatocellular carcinoma (HCC) is one of the most common and deadliest cancers worldwide with hundred thousands of deaths each year (746,000 reported deaths in 2012; Londono *et al*, 2015; Thompson *et al*, 2015). Most HCC develop as a result of chronic liver damage strongly depending on the cross talk of hepatocytes and the stromal microenvironment, which may foster a pro-inflammatory and pro-tumorigenic milieu (Coulouarn *et al*, 2012). The role of tumor stroma has long been shown to promote tumor growth and invasiveness either via direct cell–cell interaction, through the secretion of tumor-promoting cytokines such as hepatocyte growth factor (HGF) and transforming growth factor beta (TGFβ), or by modulating extracellular matrix components via integrins or fibroblast growth factors (Liu *et al*, 2011; Desert *et al*, 2016; Affo *et al*, 2017). However, recent data also suggest that the tumor stroma may exert protective anti-tumorigenic functions (Bissell & Hines, 2011; Dittmer & Leyh, 2015).

The tumor stroma consists of several cell types, including endothelial cells, macrophages, and hepatic stellate cells (HSC; Heindryckx & Gerwins, 2015). HSC and other tumor-associated mesenchymal-derived cells such as myofibroblasts and pericytes express the orphan receptor endosialin (CD248; MacFadyen *et al*, 2005, 2007; Christian *et al*, 2008; Simonavicius *et al*, 2008; Mogler *et al*, 2015). As a marker of the activated mesenchymal lineage, endosialin plays a critical role in the development and progression of liver, kidney, and pulmonary fibrosis (Chang-Panesso & Humphreys, 2015; Mogler *et al*, 2015; Bartis *et al*, 2016). In turn, endosialin negatively regulates hepatocyte proliferation, thereby balancing the epithelial and the stromal response after acute and chronic liver damage (Mogler *et al*, 2015).

Endosialin is only weakly expressed in healthy adult tissues, but prominently upregulated in the stromal compartment of

1   Division of Vascular Oncology and Metastasis, German Cancer Research Center Heidelberg (DKFZ-ZMBH Alliance), Heidelberg, Germany
2   Institute of Pathology, Heidelberg University, Heidelberg, Germany
3   Institute of Pathology, Technical University Munich, Munich, Germany
4   Department of Vascular Biology and Tumor Angiogenesis (CBTM), Medical Faculty Mannheim, Heidelberg University, Heidelberg, Germany
5   Department of Medical Physics in Radiology, German Cancer Research Center Heidelberg, Heidelberg, Germany
6   Institute of Pathology, RWTH Aachen, Aachen, Germany
7   German Cancer Consortium, Heidelberg, Germany
    *Corresponding author. Tel: +49 6221 421500; Fax: +49 6221 421515; E-mail: augustin@angiogenese.de
    †These authors contributed equally to this work

progressing tumors. Genetic inactivation or antibody-mediated inhibition of endosialin resulted in reduced primary tumor growth and metastasis in mouse models of colon cancer, melanoma, and breast cancer (Nanda *et al*, 2006; Rybinski *et al*, 2015; Viski *et al*, 2016). Endosialin therefore may be an attractive target for stromal-based therapeutic approaches in malignancies. An endosialin-blocking antibody (MORAb-004) is in early clinical development and appears to exert some clinical efficacy in a small cohort of patients with different solid extra-cerebral malignant tumors (including carcinoma, sarcoma, and neuroendocrine tumors; Diaz *et al*, 2015). We have previously observed that HSC-expressed endosialin promotes liver fibrosis and in turn negatively regulates hepatocyte proliferation, thereby acting as a balance of fibrosis vs. regeneration (Mogler *et al*, 2015). Based on these findings, the present study was aimed at studying the role of endosialin during HCC progression.

# Results and Discussion

### Endosialin is heterogeneously expressed in human HCC and inversely correlates with tumor cell proliferation

To assess the expression of endosialin during HCC progression, we performed immunohistochemical analyses in whole tissue HCC samples of different stage. Consistent with our previous findings (Mogler *et al*, 2015), expression of endosialin in healthy liver was weakly detectable exclusively in hepatic stellate cells (HSC) and portal myofibroblasts (Fig 1A). Endosialin was strongly upregulated along the fibrous septa and to lesser extent also along the sinusoids in liver cirrhosis (Fig 1B; Mogler *et al*, 2015). In dysplastic nodules (DN, *n* = 5), the premalignant lesions of HCC, endosialin was focally expressed in the sinusoidal compartment surrounding dysplastic hepatocytes (Fig 1C and Appendix Fig S1).

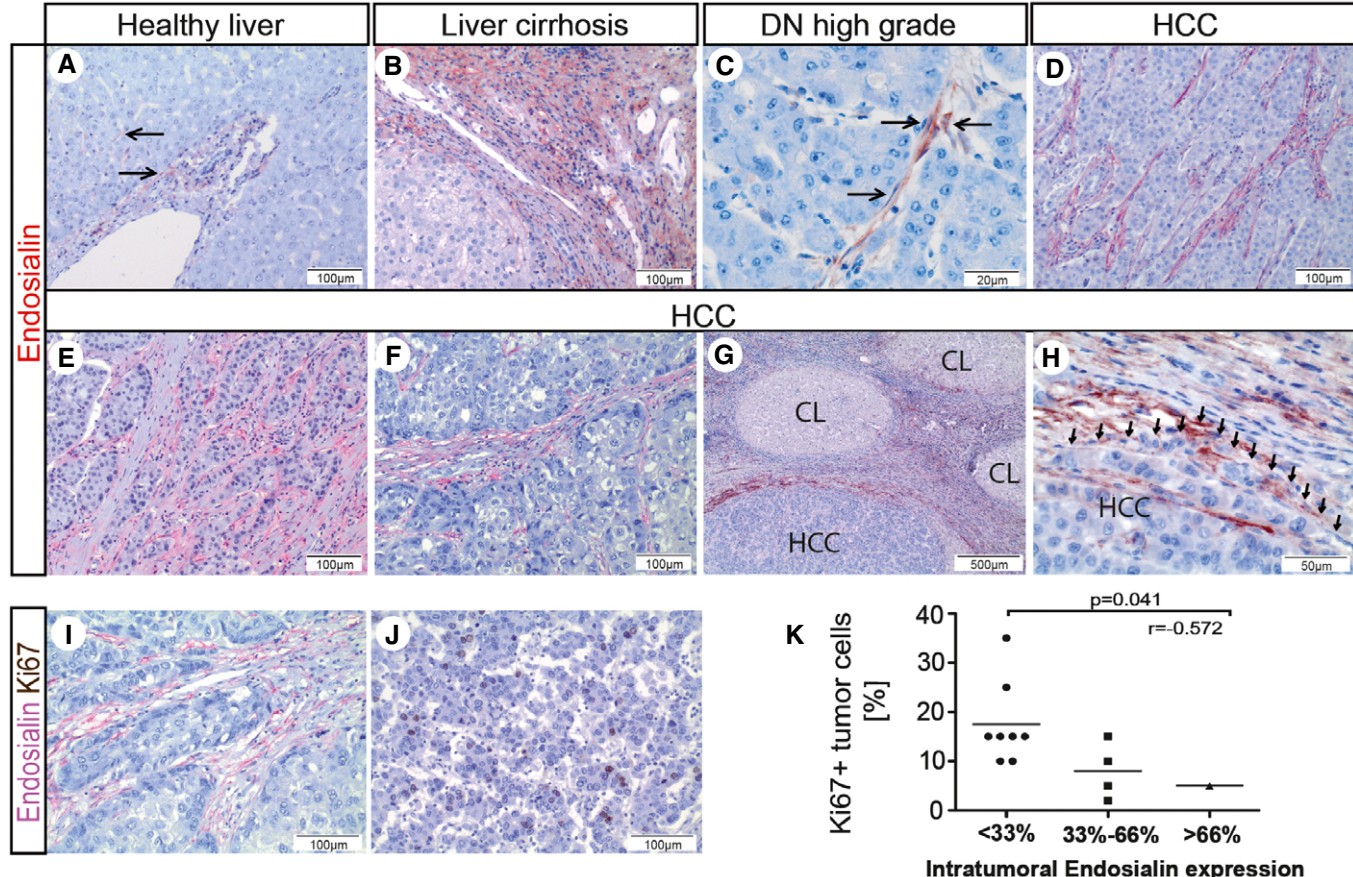

**Figure 1. Endosialin is heterogeneously expressed in human HCC and stromal-expressed endosialin inversely correlates with tumor cell proliferation.**

A–D   Endosialin immunohistochemistry staining of whole tissue slides of human healthy liver (A) (*n* = 5), cirrhotic liver (B) (*n* = 10), dysplastic nodules (DN, C) (*n* = 5), and hepatocellular carcinoma (HCC, D) (*n* = 13). Weak endosialin expression in portal tracts of normal liver (A), stronger expression of endosialin in the cirrhotic liver (B), along the sinusoids of dysplastic nodules (C), and within the HCC stroma (D).

E, F   Heterogeneous distribution of stromal-expressed endosialin.

G, H   Intense expression of endosialin at the fibrous capsule/invasion front of HCC (H shows a close-up of G; CL = nodules of cirrhotic liver).

I, J   Immunohistochemical double stains of endosialin (red) and the proliferation marker Ki67 (brown) in HCC.

K   Correlation of overall intratumoral endosialin expression and HCC tumor cell proliferation.

Data information: Scale bars: as indicated. Arrows indicate positive endosialin staining. Statistical analysis: Pearson's correlation, *r* = correlation coefficient.

In human hepatocellular carcinoma, endosialin was detectable with considerable regional variation in the stroma of all specimens with most abundant expression at the invasion front and in the fibrous capsule surrounding tumor cells ($n = 13$; Fig 1D–H). Correspondingly, when analyzing endosialin expression in tissue microarrays, only 31.5% of HCC punches ($n = 57$) showed detectable endosialin expression (Appendix Fig S2). Yet, in both, whole slides and TMA punches, endosialin expression was restricted to mesenchymal cells within the tumor (predominantly stellate cells and [myo]-fibroblasts) and tumor cells were consistently negative for endosialin. To rule out an expression of endosialin by tumor-associated endothelial cells [as proposed earlier by St Croix *et al* (2000)], double immunohistochemical stainings of CD31 and endosialin were performed confirming that endosialin in HCC was exclusively expressed by non-endothelial mesenchymal cells (Appendix Fig S3). No correlation between endosialin expression and the underlying etiology of the HCC samples was found (including viral hepatitis [$n = 5$], ASH [$n = 4$], NASH [$n = 4$]). However, when regionally quantitating tumor cell proliferation (by Ki67 immunohistochemistry of whole tissues slides), the abundance of stromal endosialin-expressing cells was inversely correlated with HCC tumor cell proliferation (Fig 1I–K).

### Enhanced HCC tumorigenesis in endosialin-deficient mice

To study the role of endosialin in an experimental model of HCC progression, we bred WT and endosialin-deficient mice ($EN^{KO}$) with mice expressing the polyoma middle T antigen Cre-inducible under the control of the albumin promoter (iAST mouse model; (Runge *et al*, 2014). $EN^{KO}$::iAST mice were viable and did not display any overt phenotype in unchallenged settings (Appendix Fig S4). Tumorigenesis in livers of WT:iAST mice and in $EN^{KO}$::iAST mice was induced by tail vein injection of Cre-expressing adenovirus, and tumor growth in WT:iAST and $EN^{KO}$::iAST mice was monitored non-invasively by weekly computed tomography (CT) scans (Fig 2A and B, and Appendix Fig S5). $EN^{KO}$::iAST presented significantly more CT-detectable tumors 6 weeks after tumor induction (Fig 2C). Tumors were harvested 8 weeks after induction, when $EN^{KO}$::iAST mice presented macroscopically significantly more tumor nodules (Fig 2D–F), higher total liver weight (Fig EV1), histologically increased tumor burden

(Fig 2G–I), and elevated tumor cell proliferation assessed by immunohistochemistry and Western blot analysis of the proliferation markers Ki67 (Fig 2J–L) and PCNA (Figs 2M–O and EV2). Histological analysis of samples harvested at earlier time points (4 weeks after tumor induction) revealed the same phenotype (Fig EV3).

### Endosialin silencing reduces HSC proliferation and enhances HCC tumor cell proliferation

We next performed cell culture experiments of HSC monocultures and HSC-HCC tumor cell co-cultures to yield mechanistic insight into the observed human and mouse phenotypes (Fig 3). Lentivirally endosialin-silenced immortalized human HSC (shEN) displayed an altered morphology with less myofibroblast-typical morphology compared to non-silenced (nsEN) control HSC (Fig 3A). Proliferation of shEN cells was strongly reduced compared to nsEN cells (Fig 3B). Stimulation of HSC by co-culture with human HCC tumor cells (Huh7) did not rescue the reduced proliferation of shEN cells (Fig 3C and D).

To study paracrine effects of HSC on HCC cells, we stimulated Huh7 cells with conditioned medium (CM) from shEN cells, which led to increased tumor cell proliferation compared to stimulation with CM from nsEN (Fig 3E). Previous experiments had identified insulin-like growth factor-2 (IGF-2) as a putative HSC-derived hepatocyte mitogen (Mogler *et al*, 2015) and a contributor to hepatocarcinogenesis (Tovar *et al*, 2010). Correspondingly, differential expression profiling experiments of shEN and nsEN LX-2 cells as well as of tumor lysates from WT:iAST and $EN^{KO}$::iAST mice revealed a significant upregulation of IGF-2 in shEN cells and in whole liver lysates of $EN^{KO}$::iAST mice (Fig 3F and G). Correspondingly, silencing the primary IGF-2 receptor, insulin-like growth factor receptor 1 (IGFR1), in cultured Huh7 cells, resulted in a significant reduction of tumor cell proliferation (Fig 3H). Albeit not formally establishing a causal relationship, the data support the hypothesis that endosialin regulates IGF-2 expression in HSC, which in a paracrine manner controls HCC tumor cell proliferation.

To test, if other paracrine factors beyond IGF-2 may contribute to the paracrine cross talk between HSC and HCC cells, we performed additional cytokine array experiments of CM from endosialin-silenced and non-silenced HSC (Appendix Fig S6A).

---

**Figure 2. Enhanced HCC tumorigenesis in endosialin-deficient mice.**

$EN^{KO}$ mice were bred into the iAST model of conditional hepatocarcinogenesis. Tumorigenesis was induced by intravenous injection of $10^9$ PFU Cre-expressing adenovirus into WT::iAST transgenic and $EN^{KO}$::iAST double-transgenic mice.

A, B    Detection of tumorigenesis by repeated CT imaging (n = nodule; st = stomach; vc = vertebral cord).
C       CT-detectable tumor nodules 6 and 7 weeks after tumor induction.
D, E    Representative images of macroscopically detectable tumor nodules in WT::iAST (D) and $EN^{KO}$::iAST (E) mice.
F       Quantitation of the mean number of macroscopically visible tumor nodules (*n* = number of nodules).
G, H    Representative histological images of tumor nodules from WT::iAST (G) and $EN^{KO}$::iAST (H) mice.
I       Quantitation of tumorous liver area in sections of livers from WT::iAST and $EN^{KO}$::iAST mice.
J, K    Representative images of Ki67 immunohistochemically stained liver sections from WT::iAST (J) and $EN^{KO}$::iAST (K) mice.
L       Quantitation of neoplastic hepatocytes in WT::iAST and $EN^{KO}$::iAST mice.
M, N    Representative images of PCNA immunohistochemically stained liver sections from WT::iAST (M) and $EN^{KO}$::iAST (N) mice.
O       Quantitation of neoplastic hepatocytes in WT::iAST and $EN^{KO}$::iAST mice.

Data information: Data are expressed as mean ± SD. All quantitative experiments have been independently reproduced two times with similar results. $n = 10$ (knockout) or $n = 12$ (wild type) mice per group and time point. Scale bars: as indicated. Statistical analysis: Student's *t*-test.

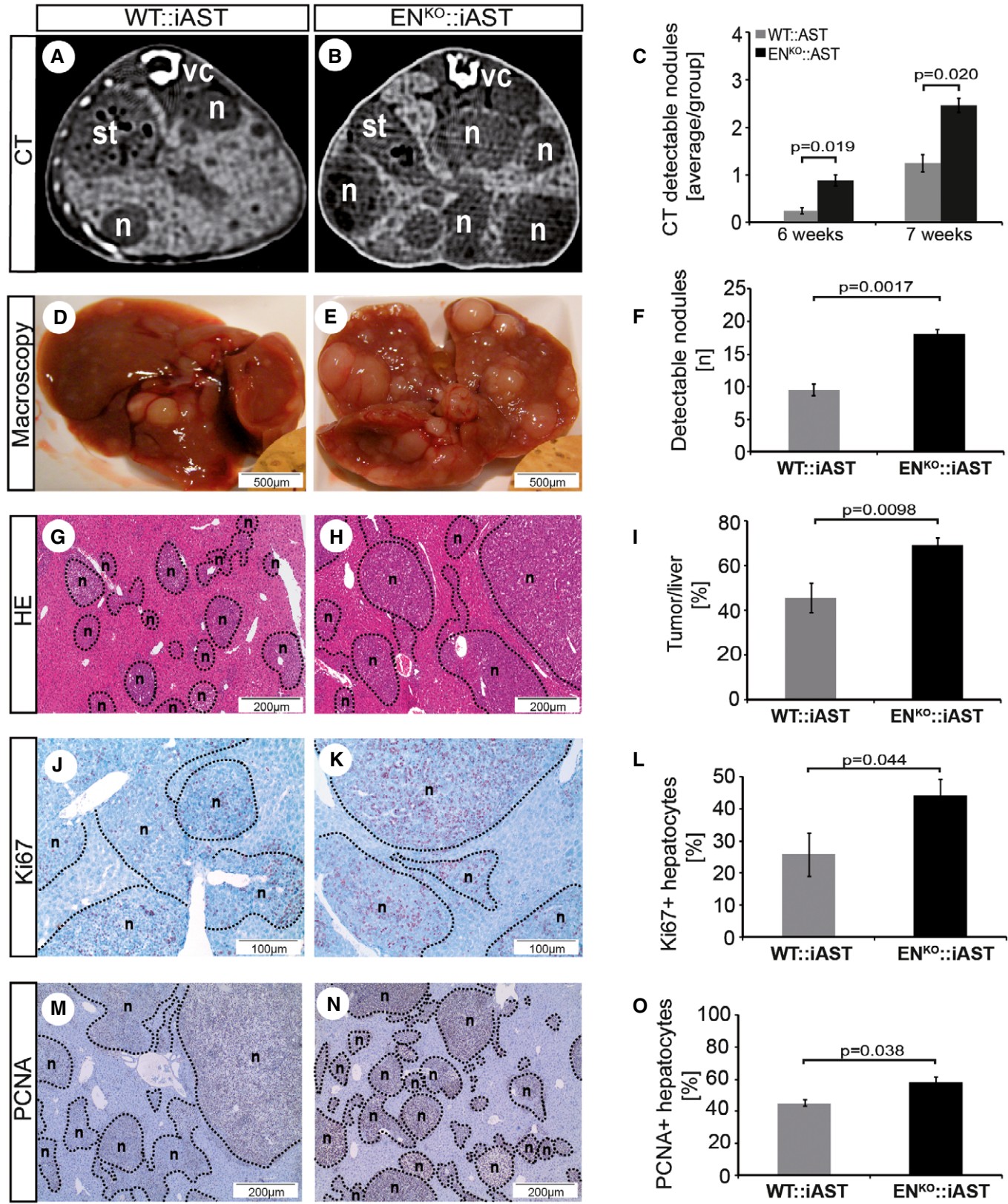

**Figure 2.**

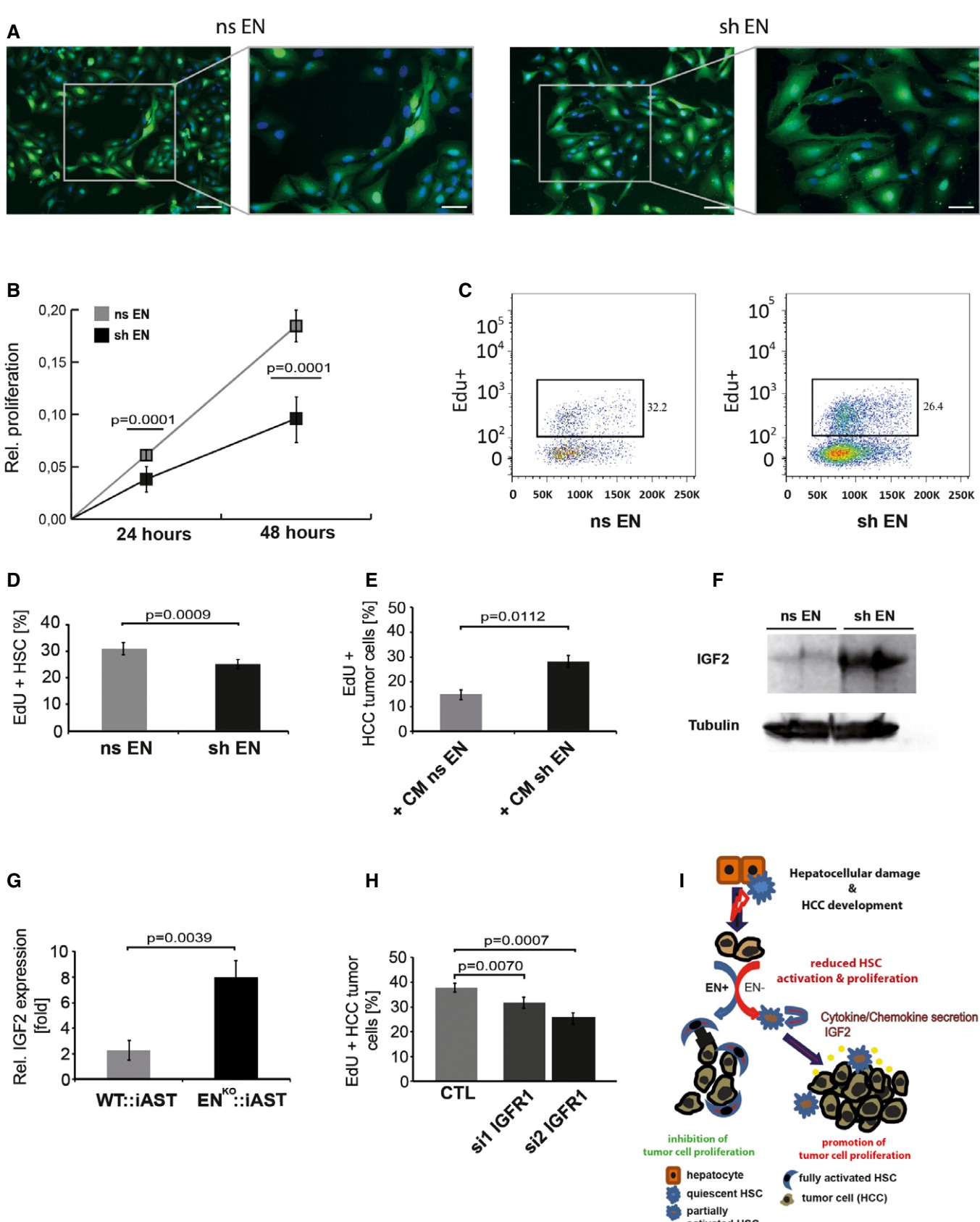

**Figure 3.**

**Figure 3. Silencing of HSC-expressed endosialin reduces HSC proliferation and enhances HCC tumor cell proliferation.**

A    Representative images of control-transduced (nsEN) or lentivirally endosialin-silenced (shEN) HSC (magnification: 10×, inserts: 20×). Counterstaining: DAPI (nuclei). Scale bars: for 10× magnification: 100 μm, for 20× magnification: 50 μm.
B    3-(4,5-Dimethylthiazol-2-yl)-2,5-diphenyltetrazoliumbromide (MTT) assay of nsEN and shEN cells.
C    EdU-based FACS analysis of nsEN and shEN HSC.
D    Quantitation of EdU-positive nsEN and shEN HSC analyzed by FACS. One representative experiment from three biological replicates.
E    Quantitation of EdU-positive Huh7 tumor cells 24 h after stimulation with conditioned media (CM) from nsEN and shEN HSC.
F    Western blot analysis of IGF-2 in nsEN and shEN HSC.
G    IGF-2 qRT–PCR of whole liver lysates from WT:: iAST and EN[KO]::iAST mice. One representative experiment from eight biological replicates.
H    Small interfering RNA (siRNA)-mediated silencing of insulin growth factor receptor 1 (IGFR1) in Huh7 tumor cells.
I    Schematic representation summarizing the findings of this study.

Data information: Data are expressed as mean ± SD. All experiments have been independently performed three times with similar results. Statistical analysis: Student's *t*-test.
Source data are available online for this figure.

These experiments identified a number of established HCC-related cytokines as being produced by HSC in an endosialin-dependent manner (see Appendix Table S1 for complete list of cytokines). Among the most strongly upregulated molecules expressed by endosialin-silenced hepatic stellate cells was the chemokine (C-C motif) ligand 5 (CCL5; Bai *et al*, 2014), retinol-binding-protein 4 (RBP4; Wang *et al*, 2011), Dickkopf-1 (DKK1; Ge *et al*, 2015), platelet-derived growth factor-A (PDGF-AA; Wei *et al*, 2014), and urokinase-type plasminogen activator receptor (uPAR; Zheng *et al*, 2014), all well-known promoters of HCC progression and aggressiveness. Huh7 cells stimulated with either CCL5 or RBP4 indeed showed higher proliferation compared to untreated tumor cells indicating that IGF2 may be a major, but not the only endosialin-controlled HSC secreted factor affecting HCC cells in a paracrine manner (Appendix Fig S6B and C). Together, these data provide strong evidence that hepatic stellate cell-expressed endosialin plays an important role in negatively controlling tumor and positively controlling stromal cell proliferation in the setting of hepatocarcinogenesis involving complex autocrine and paracrine mechanisms (Fig 3I).

The findings of the present study shed further insight into the complexity of tumor cell-stromal cell cross talk during hepatocellular carcinogenesis. Previous work on the pathogenesis of HCC has primarily focused on the molecular mechanisms governing neoplastic parenchymal cell transformation. Thereby the contribution of stromal cells to HCC initiation and progression may have been underestimated (Thompson *et al*, 2015). Among the stromal cell types, HSC have been most extensively studied for their role in regulating liver function in a paracrine manner, for example, during regeneration, fibrosis and cirrhosis from which approx. 90% of HCC develop (Forner *et al*, 2012). Yet, the mechanisms of HSC contribution to HCC represent a complex, to date poorly understood role in tumorigenesis. Intriguing findings are highly contradictory implicating a "*good*" vs. "*bad*" phenotype of HSC. For example, the activation and associated phenotype of HSC have been shown to contribute to HCC development and progression by secreting proliferation-inducing cytokines, such as hepatocyte growth factor (HGF; Matsumoto & Nakamura, 2006) or by initiating in a paracrine manner tumor angiogenesis (Zhu *et al*, 2015). In turn, activated HSC are a source of transforming growth factor beta (TGFβ), which exerts growth inhibitory effects, particularly in early tumor stages (Meindl-Beinker *et al*, 2012). These apparently discrepant findings may point toward

context-dependent pro- and anti-tumorigenic effects of HCC-associated HSC. In the present study, we could show in definite genetic settings that endosialin, which is in the liver expressed by HSC and stromal myofibroblasts, acts as a negative regulator of HCC. Moreover, albeit restricted to a limited cohort of samples, there was a strong inverse correlation between tumor cell proliferation and stromal endosialin expression in human HCC suggesting that the paracrine cross talk observed in the preclinical mouse model may also exist in human HCC. The loss of HSC-expressed endosialin led to an increase of tumor-promoting secreted factors, including IGF-2 and CCL5.

The findings of this study also contribute to the increasing appreciation that stromal cells do not just exert pro-tumorigenic functions, but may restrain tumor growth. For example, preclinical pancreatic ductal adenocarcinoma experiments have suggested that the targeting of tumor-associated fibroblasts may stimulate rather than inhibit tumor growth (Ozdemir *et al*, 2014; Rhim *et al*, 2014). These landmark studies have inspired the concept that stromal reprogramming may be therapeutically exploited to balance tumor cell proliferation (Rowley, 2014). Our findings expand these concepts by ascribing a distinct cell surface receptor, the orphan receptor endosialin (CD248), to the paracrine tumor growth regulating properties of HCC-associated hepatic stellate cells. Future work will need to focus on the mechanisms controlling the activation status of HSC that determines the phenotypic switch from "*good*" to "*bad*" (or vice versa?). As shown here, while endosialin-expressing and fully activated HSC impair tumor growth, endosialin-deficient, partially activated HSC promote HCC progression. Taken together, this study establishes a causal inverse relationship between hepatic stellate cell-expressed endosialin and growth of HCC and put a cautionary note on the potential application of endosialin-blocking antibodies in HCC.

# Materials and Methods

### Patient samples

This study was registered at the tissue bank of the National Center for Tumor Diseases (NCT, Heidelberg, Germany) and performed according to the Declaration of Helsinki; written informed consent was obtained from all patients. All patient specimens were exclusively provided in a pseudonymized form

    

according to the Standard Operating Procedures of the NCT, approved by the Ethics Committee of the Heidelberg University (Ethical votes # 206/207, 2005).

## Animals

All animal experiments were performed according to the guidelines of the local Animal Use and Care Committees and approved by the Regierungspräsidium in Karlsruhe (35-9185.81/G-228/10). Animals (C57/Bl6, wild type (WT::iAST) or endosialin knockout (EN^KO^:: iAST)) were housed in barriers at the animal facility of the DKFZ with free admission to food and water. Sex- and age-matched endosialin WT::iAST and EN^KO^::iAST C57/Bl6 mice were used for tumor experiments. Adeno-Cre virus (100 µl; Vector Biolabs) was diluted to $1 \times 10^9$ PFU/100 µl with isotonic 0.9% NaCl and injected into the tail vein.

## Computer tomography (CT)

Mice were anaesthetized using isoflurane (1.5%) and oxygen (0.5 l/min). Respiratory-gated volumetric computed tomograph (VCT) imaging was done on a prototype flat-panel equipped volumetric computed tomograph (Volume CT, Siemens; tube voltage, 80 kV; tube current, 50 mA; scan time, 80 s; frames per second, 120; rotation speed, 10 s; Kernel H80a; matrix $512 \times 512$. Fenestra LC contrast agent (100 µl; LC-133, Art, Canada) was injected in the tail vein. Images were processed with the program OsiriX.

## Processing of formalin-fixed paraffin-embedded and cryo-sectioned tissues

All organs and tissues were fixed in 4% PFA overnight, processed and stained (H&E) according to standard procedures. For cryofixation, tissues were placed in OTC cryo-medium, or alternatively on cork plates, and were snap-frozen in liquid nitrogen or on dry ice.

## Immunohistochemistry stainings on paraffin sections

Paraffin sections were dewaxed according to standard procedures. For antigen-retrieval slides were boiled for 20 min in 0.01 M pH 6.0 citrate target retrieval buffer and treated with 3% $H_2O_2$ for 15 min to block endogenous peroxidase. Slides were incubated with primary antibodies [endosialin: kindly provided by C. Isacke 1:500; Ki67 (human): 1:100 (Dako, Hamburg, Germany); Ki67 (mouse): 1:50 (Dako, Hamburg, Germany), PCNA: 1:500 (My Biosource, San Diego, USA)]. Detection was performed via a biotin–peroxidase complex according to manufacturer's protocol (Dako, Hamburg, Germany). Sections were counterstained with hematoxylin.

## Quantitative real-time PCR (qRT–PCR)

RNA from cells or pieces of liver was purified by the Rneasy Mini Kit from Qiagen according to manufacturer's instructions. Concentration and purity were measured on the Nanodrop (Eppendorf). cDNA was generated using the Quantitect® Reverse Transcription

Kit (Qiagen) according to manufacturer's instructions. Quantitative real-time RT–PCR (qRT–PCR) was performed on an ABI StepOne-Plus cycler using TaqMan probe sets purchased from Applied Biosystems.

## Western blot analysis

Protein lysates were prepared from frozen organs or cryo-material. Liver material was lysed in NP-40 lysis buffer with phosphatase inhibitor orthovanadate (2 mM). Protein concentrations were determined by Bradford or the BCA assay. Proteins were separated on 10% polyacrylamide/SDS gels, blotted on nitrocellulose membranes, and incubated with primary antibodies at 4°C [PCNA: 1:500 (My Biosource, San Diego, USA)] overnight. Detection of chemiluminescence was done with ECL Western blotting substrate (Pierce). Scanning and evaluation was performed using the Amersham Western Blotting System (GE Healthcare, Munich, Germany).

## Cells and SH mediated lentiviral knockdown

The human HSC cells LX-2 were purchased from Millipore (Darmstadt, Germany). LX-2 were cultured in DMEM 1 g glucose, 2% fetal calf serum, and 1% penicillin/streptomycin (PS), at 37°C, 5% $CO_2$. Cells were passaged as necessary. Endosialin knockdown was achieved by GFP-labeled lentiviral-mediated transfection of LX-2 cells using either non-silencing (nsEN) or endosialin-silencing (shEN) virus as already published (Mogler *et al*, 2015). Silencing of gene expression was validated by qPCR and was consistently above 90%. Huh7 cells were kindly provided from PD Dr. K. Breuhahn (Institute of Pathology, Heidelberg) and cultured in DMEM (4.5 g glucose, 10% FCS, 1% PS).

## MTT assay

Of 2,000 LX-2 cells (nsEN and shEN) were seeded in triplicates into 96-well plates ($T = 0, 24, 48$ h). Assay was performed according to manufacturer's protocol (Roche, Basel, Switzerland).

## siRNA transfection

Of 70,000 Huh7 cells were plated in 6-well plates. Solution A (10 µl siNTC (Ambion) or siIGF1R_ds1 (=si 1) or siIGF1R_ds3 (=si 2; Eurofins Genomics) in 100 µl OptiMEM (Gibco) was incubated for 10 min at RT. Solution B (5 µl Oligofectamine (Invitrogen) in 100 µl OptiMEM, incubated 10 min at RT) was mixed with Solution A and incubated for 30 min at RT. The siRNA mix (A + B) was added dropwise to the cells and incubated for 4 h at 37°C. Knockdown was checked after 48 h by qRT–PCR.
IGF1R ds1 (GGACUCAGUACGCCGUUUA)TT
IGF1R ds3 (GGCCAGAAAUGGAGAAUAA)TT

## EdU assay

EdU assay was used for both Huh7 and HSC cells (60,000 Huh7 and 40,000 LX-2/well with three wells per experimental condition). Cells were cultured in 2% FCS at least 24 h prior to experimental procedure. EdU intake and staining was performed using the Click-iT®

EdU Alexa Fluor® 647 Imaging Kit according to the manufacturer's protocol (Thermo Fisher Scientific, Munich, Germany). Ten random pictures of each well and condition were taken. Pictures were analyzed by two independent investigators (C.M. and C.K.) using ImageJ Software (FIJI).

For stimulation experiments, cytokines (CCL5 and RBP4) were added to the medium (10 ng/ml for CCL5 and 50 ng/ml for RBP4) for 24 h.

### EdU FACS (co-cultures Huh7 and Lx-2)

60,000 Huh7 and 40,000 LX-2 (± shEN) cells were incubated for 48 h prior to EdU (1:1,000) incubation for 3 h. EdU staining was performed according to manufacturer's protocol (Thermo Fischer Scientific). Prior to FACS analysis, cells were stained with FxCycle and resuspended in 100 μl PBS.

### Cytokine array

Human XL cytokine array was purchased from R&D, and level of cytokines was assessed as per directions of the manufacturer. Briefly, the array membranes were blocked with blocking buffer at room temperature and incubated with supernatant derived from co-cultures (Huh7 and LX-2 ± shEN) overnight. Detection antibody cocktail was added to the membranes. After incubation and washing, streptavidin–HRP was added to each membrane. After incubation and washing, the cytokines were detected by chemiluminescence reaction. Spots were quantified using FIJI software after background subtraction. Normalized data were analyzed for endosialin knockdown-specific results and used to determine the mean differences in cytokine abundance.

### Statistics

All results are expressed as mean ± SD. Differences between WT:: iAST and EN^KO::iAST mice groups were analyzed using the two-tailed unpaired Student's $t$-test. The statistical difference between different time points in WT mice was analyzed using the two-tailed paired or unpaired $t$-test. Correlations were analyzed using the Pearson (bivariate) correlation. Differences $P \leq 0.05$ were considered statistically significant.

Expanded View for this article is available online.

### Acknowledgements
This work was funded by grants from the Deutsche Forschungsgemeinschaft (SFB-TR23 "Vascular Differentiation and Remodeling" [to HGA] and SFB873 "Maintenance and Differentiation of Stem Cells in Development and Disease" [to HGA]) and the Helmholtz Alliance "Preclinical Comprehensive Cancer Center" [to HGA].

### Author contributions
CM, CK, and MW designed research; CM, CK, MW, AR, EB, and DK performed experiments; CM, CK, MW, AR, DK, TL, and PS analyzed data; and CM and HGA wrote the manuscript. All authors read and approved the manuscript.

### Conflict of interest
The authors declare that they have no conflict of interest.

## The paper explained

### Problem
We have previously shown that hepatic stellate cells (HSC) promote hepatocyte proliferation during liver regeneration through the orphan receptor endosialin (CD248). We hypothesized that a growth promoting effect during tissue regeneration could possibly translate into a tumor-promoting effect in hepatocellular carcinoma (HCC), one of the deadliest cancers worldwide. Employing a genetic model of endosialin deficiency and a genetic model of HCC tumorigenesis, we studied the role of endosialin during the growth of HCC.

### Results
Genetic inactivation of endosialin resulted in accelerated tumor growth in an inducible mouse model of HCC. Several hepatotropic cytokines, including IGF2, RBP4, DKK1, and CCL5, very identified as being negatively regulated by endosialin. Histological analysis of human HCC samples revealed an inverse correlation between tumor cell proliferation and stromal endosialin expression.

### Impact
The experiments demonstrate that HCC-associated hepatic stellate cells are capable to inhibit tumor growth and progression through the orphan receptor endosialin. The data contribute to the emerging theme that stromal cells do not just promote tumor growth, but may in fact be part of the host's defense aimed at restraining tumor growth.

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
