## [Review Process File · EMBO Molecular Medicine]

Hepatic stellate cells limit hepatocellular carcinoma progression through the orphan receptor endosialin

Carolin Mogler, Courtney König, Matthias Wieland, Anja Runge, Eva Besemfelder, Dorde Komljenovic, Thomas Longerich, Peter Schirmacher, and Hellmut G. Augustin

Corresponding author: Hellmut Augustin, Medical Faculty Mannheim, University of Heidelberg, and German Cancer Research Center Heidelberg

Review timeline:

Submission date:	19 October 2016
Editorial Decision:	22 November 2016
Revision received:	27 February 2017
Editorial Decision:	13 March 2017
Revision received:	13 March 2017
Accepted:	15 March 2017

Transaction Report:

Editor: Roberto Buccione

1st Editorial Decision

22 November 2016

Thank you for the submission of your manuscript to EMBO Molecular Medicine. We have now heard back from the three reviewers whom we asked to evaluate your manuscript.

As you will see the three reviewers are quite positive although they do raise some concerns on your manuscript, which I would basically summarise as follows. Many comments are focused on improving presentation of data and providing much better experimental details, both very important aspects in our quest at EMBO Molecular Medicine to improve the reproducibility of published findings. Reviewer 2, instead notes that your claim that IGF2 is a major mediator of endosialin loss in HSCs is not fully supported by the data and would like you to attempt further experimentation to solidify this aspect. S/he also has a concern on the cytokine array experiment and suggests validating some key cytokines.

In conclusion, while publication of the paper cannot be considered at this stage, we would be pleased to consider a revised submission, with the understanding that the Reviewers' concerns must be addressed including further experimentation as mentioned above. Eventual acceptance of the manuscript will entail a second round of review.

Reviewer 1 also suggests the reorganisation of your manuscript as a Research Article rather than a Report. I would not necessarily ask for this but if you feel that full compliance with the numerous requests would justify this transformation, I would have no objection. Needless to say, I would in such case suggest you to upgrade a couple of the most relevant figures from appendix to main.

I look forward to seeing a revised form of your manuscript as soon as possible.

***** Reviewer's comments *****

Referee #1 (Comments on Novelty/Model System):

All results are potentially interesting and novel. However there are a few major issues that need to be addressed before the paper is fully convincing.

Referee #1 (Remarks):

The manuscript by Mogler et al. aimed to investigate the role of hepatic stellate cell-expressed endosialin in cell proliferation during hepatocarcinogenesis. It is a very concisely written paper and presents results in a logical order, highlighting a negative correlation of endosialin expression and HCC tumour cell proliferation. The presented results are convincing, however sometimes the manuscript would have benefited from more background information or additional experiments to round off the story.

Comments:

Introduction: Very brief and a lot more background could be provided with regards to HCC tumour niche development and cellular crosstalk of key regulators to set the scene for endosialin and the rest of the manuscript.

Results:

Figure 1: The authors show immunohistochemical stainings of human liver biopsy material for endosialin and Ki67. No patient information is provided. What is the chronic liver injury background that the HCCs developed on? Viral hepatitis, alcoholic liver disease, non-alcoholic fatty liver disease? Different aetiologies may induce divergent histological features and contribute to the considerable regional variations that the authors are describing. It is stated that only hepatic stellate cells and portal myofibroblasts express endosialin but it is not possible to derive this from the shown images. Dual immunofluorescence for endosialin and respective cell markers would be more convincing since it cannot be ruled out that other cell types in the portal niche induce endosialin under appropriate conditions (such as liver progenitor cells for example).

Figure 2: Methods details for PCNA staining are missing. Manufacturer details are missing for detection with the biotin-peroxidase complex. Western Blot primary antibody details missing. Q is difficult to interpret due to a lack of labelling and the result shown for background fluorescence in between the two experimental groups.

Figure 3: The authors describe an altered morphology in HSCs when endosialin was silenced, however Panel A is difficult to interpret due to suboptimal quality and no higher magnification insert provided. Also, in the text the altered morphology is referred to in Figure 3A and B, however B shows the proliferation rate, not the morphology as such. There are not enough details provided for the EDU assay and even with the manufacturer's details, it is not possible to find the correct assay and potentially replicate the results. For LX2 culture it is described that the cells were either grown in 2% or 10% FBS, which is a big difference but no details are provided as to which concentrations were chosen for which assay.

Overall, this work has a lot of potential and the results are potentially exiting but the manuscript feels preliminary due to the lack of provided details and lack of discussion.

I do believe that this report would be better suited as a full research article to be able to provide more background as well as experimental details and make the results more convincing.

Referee #2 (Comments on Novelty/Model System):

the manuscript uses a mouse model recently reported by their lab, cell lines and studies of human tissues. This is very suitable and the claims are supported by the data.

Referee #2 (Remarks):

Endosialin is an orphan receptor that is expressed in the liver in disease states, mostly in hepatic stellate cells (HSCs). These cells are known to play important roles fibrosis and in regulating liver regeneration. However, their role in HCC is not clear and they were shown to play both positive and negative roles. The mechanisms that impart such negative or positive roles on HSCs are not known.

Mogler et al have previously shown that endosialin is an important regulator of HSCs in liver fibrosis and regeneration. They now go on to reveal that endosialin is a key molecule that imparts an anti-oncogenic role on HSCs. The article reveals that loss of endosialin changes the phenotype of HSCs, and they assume a phenotype characterized by secretion of several pro-proliferative cytokines that are known to be important for HCC progression. The cell line and well conducted mouse studies are accompanied by a convincing set of human data that goes along with their claim.

Comments:

In my mind the phenotypic switch (i.e. that HSCs can assume "good" vs. "bad" phenotype) should be made more prominent in the abstract, discussion as well as the graphic abstract in fig 3J.

On the other hand, the authors seem to claim that the data showing that silencing IGFR1 in hepatocytes (Fig 3H) results in reduced HCC proliferation proves that IGF2 is a major mediator of the effect of endosialin loss in HSCs. While the data provides correlation it does not prove this. To do so, they should overexpress IGF2 in endosialin sh cells, and use those in co culture. I am not sure this experiment will work, as there are many other mediators. The bottom line - the conclusions should be softened (i.e. "indicating" changed to "supporting the possibility" or the like).

The cytokine array experiment (fig 3I) is very important as it shows a real phenotype switch in HSCs upon endosialin knockdown. It is not clear how many times this was repeated. If only once, then some of the highly differentially regulated cytokines should be assessed in an additional manner (specific ELISA, real time PCR if known to be regulated at mRNA level etc.).

It is not clear if the data presented in fig 2a-c is statistically significant.

Referee #3 (Remarks):

This is an interesting and well-executed study that reports on the functional contribution of hepatic stellate cells (HSC) derived endosialin expression in hepatocarcinogenesis. Human hepatocellular carcinoma (HCC) tissues were used to establish the correlation between HSC derived endosialin expression and tumor progression. The authors also used genetically engineered endosialin KO mice to test their hypothesis: HSC endosialin promote hepatocyte proliferation, implicating IGF2 paracrine signaling. Finally, a knock down culture system was used to define the communication between HCC and HSC. The study is well constructed and thoroughly tested. Pending minor clarifications, this manuscript thus offers a novel insight on hepatocarcinogenesis.

1. The authors previously reported that HSC derived endosialin is implicated in liver fibrosis and regeneration (Mogler et. al., 2015). This should be better emphasized in the introduction of the manuscript as highly relevant for the hypothesis testing brought forth in this manuscript.
2. Figure 1, panel C, the staining is difficult to appreciate; please consider presenting a higher magnification image. The figure legend lettering appears to be incorrect, please verify.
3. In Extended view Fig. 1, are the authors implying that there are heterogeneous expression pattern/level on TMA? Have they correlated these differences in expression with respect to HSC content in TMA sections?
4. Figure 2C: are these differences statistically different? For the data presented in Figure 2, The number of animals used in each group is not listed in the figure legend.
5. What is the meaning of figure 2P-Q? Is it supportive of the data presented in M-O?
6. Figure 3, please indicate the percent down regulation of endosialin in the knockdown strategy?
7. Figure 3E. Is there a difference in % Edu+ HCC tumor cells between none-treated (Just HCC tumor cells) vs treated with CM ns EN? Please note the poor quality of the image in panel 1 renders it difficult to appreciate the results.

Response to Reviewers' comments

Reviewer #1

COMMENT ON NOVELTY/MODEL SYSTEM: All results are potentially interesting and novel. However, there are a few major issues that need to be addressed before the paper is fully convincing.

RESPONSE TO NOVELTY/MODEL SYSTEM: We sincerely appreciate the reviewer's overall positive assessment of our work and would like to thank him/her for the thoughtful suggestions to further advance the manuscript. All comments of the reviewer have been addressed in additional experiments and editorial revisions. The changes to the manuscript are outlined in detail below.

GENERAL COMMENT: The manuscript by Mogler et al. aimed to investigate the role of hepatic stellate cell-expressed endosialin in cell proliferation during hepatocarcinogenesis. It is a very concisely written paper and presents results in a logical order, highlighting a negative correlation of endosialin expression and HCC tumor cell proliferation. The presented results are convincing, however sometimes the manuscript would have benefited from more background information or additional experiments to round off the story.

RESPONSE TO GENERAL COMMENT: We have redrafted the introduction and discussion sections to include more background information. Following the reviewer's suggestions, we have performed additional experiments that are now included in the main figure and the supplemental material of the revised manuscript and outlined in detail below.

COMMENT 1: Introduction: Very brief and a lot more background could be provided with regards to HCC tumor niche development and cellular crosstalk of key regulators to set the scene for endosialin and the rest of the manuscript.

RESPONSE 1: The '*Brief Report*' format limits the overall length of the manuscript. However, by expanding the introduction and discussion, we have aimed at addressing the reviewer's concerns.

COMMENT 2: Figure 1: The authors show immunohistochemical stainings of human liver biopsy material for endosialin and Ki67. No patient information is provided. What is the chronic liver injury background that the HCCs developed on? Viral hepatitis, alcoholic liver disease, non-alcoholic fatty liver disease? Different etiologies may induce divergent histological features and contribute to the considerable regional variations that the authors are describing. It is stated that only hepatic stellate cells and portal myofibroblasts express endosialin but it is not possible to derive this from the shown images. Dual immunofluorescence for endosialin and respective cell markers would be more convincing since it cannot be ruled out that other cell types in the portal niche induce endosialin under appropriate conditions (such as liver progenitor cells for example).

RESPONSE 2: Following the reviewer's suggestion, we compared the amount of Endosialin to the corresponding clinical data for the samples used for this study. We do not see a correlation with regard to the etiology of the HCC samples (13 whole slide samples of HCC including 5 cases of viral hepatitis [4x HBV, 1x HCV], alcoholic steatohepatitis/ASH [4 cases] or non-alcoholic steatohepatitis/NASH [4 cases]). To exclude negative data resulting from a small sample number, we increased the number of HCC samples including another cohort of HCC samples (20 samples of fresh frozen HCC samples: 7x viral hepatitis [4x HBV/3x HCV], 7x ASH, 5x NASH, 1x unknown). Again, Endosialin levels were independent of the etiology. We included this in the results section of the revised manuscript (page 5). We interpret these findings (also in view of our previous work on Endosialin expression during liver fibrosis [Mogler et al., EMBO Mol Med, 2015] as indicating that activation of hepatic stellate cells and increase of Endosialin expression seems to be independent of the cause of damage: Any kind of acute or chronic liver injury will lead to an activation of hepatic stellate cells which is accompanied by Endosialin expression. Moreover, chronic liver damage is often multifactorial: The main cause of liver damage might be aggravated by additional risk factors (e.g. patients with history of chronic alcohol abuse might suffer from metabolic syndrome as well or e.g. patients with history of chronic viral hepatitis might experience additional drug induced liver injury [DILI]).

Addressing the second point of the reviewer: We have previously analyzed in substantial detail

Endosialin expression in the normal and fibrotic liver (Mogler et al., EMBO Mol Med, 2015). We have in this study analyzed gene expression in isolated liver cell populations (qPCR, IHC, IF) and could robustly trace Endosialin expression to cells from the mesenchymal lineage with quantitatively highly preferential expression by hepatic stellate cells and portal (myo-)fibroblasts. We have not yet performed Endosialin genetic fate mapping experiments and can therefore not formally exclude that some smaller progenitor cell populations could express Endosialin. Yet, quantitatively speaking, we are confident that hepatic stellate cells and portal (myo-)fibroblasts are the primary Endosialin expressing cells in the liver. To accommodate the reviewer's concern, we have employed some more cautionary wording ('*predominantly*') in the manuscript (page 4).

COMMENT 3: Figure 2: Methods details for PCNA staining are missing. Manufacturer details are missing for detection with the biotin-peroxidase complex. Western Blot primary antibody details are missing. Q is difficult to interpret due to a lack of labelling and the result shown for background fluorescence in between the two experimental groups.

RESPONSE 3: We included the missing information (revised methods section in the expanded view material). Concerning the Western blot analysis shown in Fig. 2Q of the original manuscript, we apologize for the misleading representation of the data in the original manuscript. We used the GE Healthcare Amersham Western Blot machine (panel on the left in Fig. 2Q), which automatically quantifies the amount of detected target protein in relation to total protein. This automatic quantification was shown on the right side of Fig. 2Q. The data are supportive of the immunohistochemical images shown in 2M and 2N. We therefore transferred these data to the supplemental material and changed the arrangement (quantitation below bands [expanded view Fig. 7]).

COMMENT 4: Figure 3: The authors describe an altered morphology in HSCs when endosialin was silenced. However, panel A is difficult to interpret due to suboptimal quality and no higher magnification insert provided. Also, in the text the altered morphology is referred to in Figure 3A and B. However, B shows the proliferation rate, not the morphology as such. There are not enough details provided for the EdU assay and even with the manufacturer's details, it is not possible to find the correct assay and potentially replicate the results. For LX2 culture it is described that the cells were either grown in 2% or 10% FBS, which is a big difference but no details are provided as to which concentrations were chosen for which assay.

RESPONSE 4: We have replaced the images with higher quality images to show Endosialin-silenced and mock-transfected LX2 cells. Following the reviewer's suggestions, we also included a higher magnification image in the main figure (Fig. 3A of revised manuscript). The altered morphology in Endosialin-silenced cells vs. mock-transfected cells includes more abundant cytoplasm and slightly enlarged nuclei (included in revised manuscript). As suggested, additional detail on the EdU assay has been included in the method section (revised expanded view method section). The EdU assays are a reliable surrogate marker to assess the proliferation of cells (HSC and Huh7) in vitro (e.g. as previously shown in Mogler et al., EMBO Mol Med, 2015 or Hu et al., Science, 2014). All experiments have been performed using 2% FCS and performed 3 times independently with 3 biological replicates for each experimental condition (included in methods section). After performing the EdU staining protocol according to the manufacturer's protocol, ten images per well and condition were randomly taken. Images were then automatically quantified by two different investigators (C.M. & C.K.) using Image J software (FIJI).

COMMENT 5: Overall, this work has a lot of potential and the results are potentially exciting but the manuscript feels preliminary due to the lack of provided details and lack of discussion.

RESPONSE 5: Expanding the manuscript by addressing the reviewer's very helpful suggestions leading to the inclusion of the results of additional experiments and editorial revision of manuscript, we believe that the manuscript has substantially grown and will now make a strong contribution.

COMMENT 6: I do believe that this report would be better suited as a full research article to be able to provide more background as well as experimental details and make the results more convincing.

RESPONSE 6: We sincerely appreciate the reviewer's suggestion to expand this manuscript towards a full research article. The work is timely, competitive and has the potential to conceptually advance

the field. We would consequently like to communicate the findings rather earlier than later. We therefore opted for the '*Brief Report*' format which we consider quite suitable towards this end. We do believe that expansion of the manuscript during the revision have significantly strengthened the study to now make it a "round story".

Reviewer #2

COMMENT ON NOVELTY/MODEL SYSTEM: The manuscript uses a mouse model recently reported by their lab, cell lines and studies of human tissues. This is very suitable and the claims are supported by the data.

RESPONSE TO NOVELTY/MODEL SYSTEM: We sincerely thank the reviewer for his/her very positive feedback.

GENERAL COMMENT: Endosialin is an orphan receptor that is expressed in the liver in disease states, mostly in hepatic stellate cells (HSCs). These cells are known to play important roles fibrosis and in regulating liver regeneration. However, their role in HCC is not clear and they were shown to play both positive and negative roles. The mechanisms that impart such negative or positive roles on HSCs are not known. Mogler et al have previously shown that endosialin is an important regulator of HSCs in liver fibrosis and regeneration. They now go on to reveal that endosialin is a key molecule that imparts an anti-oncogenic role on HSCs. The article reveals that loss of endosialin changes the phenotype of HSCs, and they assume a phenotype characterized by secretion of several pro-proliferative cytokines that are known to be important for HCC progression. The cell line and well conducted mouse studies are accompanied by a convincing set of human data that goes along with their claim.

RESPONSE TO GENERAL COMMENT: We again would like to thank the reviewer for his/her positive assessment of our work.

COMMENT 1: In my mind the phenotypic switch (i.e. that HSCs can assume "good" vs. "bad" phenotype) should be made more prominent in the abstract, discussion as well as the graphic abstract in fig 3J.

RESPONSE 1: We thank the reviewer for his/her suggestion with which we fully agree. We have highlighted this issue and expanded the introduction and discussion of the revised manuscript accordingly.

COMMENT 2: On the other hand, the authors seem to claim that the data showing that silencing IGF2 in hepatocytes (Fig 3H) results in reduced HCC proliferation proves that IGF2 is a major mediator of the effect of endosialin loss in HSCs. While the data provides correlation it does not prove this. To do so, they should overexpress IGF2 in endosialin sh cells, and use those in co culture. I am not sure this experiment will work, as there are many other mediators. The bottom line - the conclusions should be softened (i.e. "indicating" changed to "supporting the possibility" or the like).

RESPONSE 2: We fully agree with the reviewer that the correlative finding is not sufficient to make a causal claim. Endosialin-silenced HSC produce more IGF2, which we hypothesize drives HCC proliferation. As such, an IGF2 gain-of-function experiment would not be reasonable. Yet, to accommodate the reviewer's concern, we have redrafted the text accordingly and used more cautionary wording.

COMMENT 3: The cytokine array experiment (Fig 3I) is very important as it shows a real phenotype switch in HSCs upon endosialin knockdown. It is not clear how many times this was repeated. If only once, then some of the highly differentially regulated cytokines should be assessed in an additional manner (specific ELISA, real time PCR if known to be regulated at mRNA level etc.).

RESPONSE 3: We included the cytokine array experiment to highlight the observation that IGF2 may be a primary mediator of the HSC Endosialin effects on hepatocarcinogenesis, but probably not the only one. The array experiment was performed several times with various passages of HSC. Depending on the HSC passage, there is some variation in the array results and the hierarchy of the

most regulated cytokines. We therefore performed additional EdU stimulation experiments with Huh7 tumor cells for the most robustly regulated cytokines identified in the arrays, namely CCI5 and RBP4, for which pro-proliferative effects of both, CCI5 and RBP4 have previously been reported for HCC cells. These experiments essentially identified the same proliferative effect as observed for IGF2 (included in expanded view figure 11). The text has been correspondingly rephrased to explicitly state that IGF2 may be a major, but not the only contributor to the observed effects.

COMMENT 4: It is not clear if the data presented in fig 2a-c is statistically significant.

RESPONSE 4: The data presented in Fig. 2C showed the percentage of mice presenting with CT-detectable tumor nodules. The data (as indicated) are indeed not statistically significant (using Fisher's test: $p=0.084$ for 6 weeks and $p=0.096$ for 7 weeks). This graph was supposed to support the hypothesis that loss of Endosialin plays a role in early tumor development. Yet, we agree with the reviewer that the simple tumor / no tumor analysis is too crude for a meaningful quantitation. We have therefore quantitated the number of CT-detectable nodules $>0.5\text{ml}$, which yields meaningful readouts and demonstrates the highly significant effect of Endosialin absence or presence on tumor growth (Fig. 2C of revised manuscript).

Reviewer #3

GENERAL COMMENT: This is an interesting and well-executed study that reports on the functional contribution of hepatic stellate cells (HSC) derived endosialin expression in hepatocarcinogenesis. Human hepatocellular carcinoma (HCC) tissues were used to establish the correlation between HSC derived endosialin expression and tumor progression. The authors also used genetically engineered endosialin KO mice to test their hypothesis: HSC endosialin promote hepatocyte proliferation, implicating IGF2 paracrine signaling. Finally, a knock down culture system was used to define the communication between HCC and HSC. The study is well constructed and thoroughly tested. Pending minor clarifications, this manuscript thus offers a novel insight on hepatocarcinogenesis.

RESPONSE TO GENERAL COMMENT: We sincerely thank the reviewer for his/her very positive overall feedback.

COMMENT 1: The authors previously reported that HSC derived endosialin is implicated in liver fibrosis and regeneration (Mogler et. al., 2015). This should be better emphasized in the introduction of the manuscript as highly relevant for the hypothesis testing brought forth in this manuscript.

RESPONSE 1: We have expanded the introduction to better highlight the results from our previous work, which have led to the experimental hypothesis of the current manuscript.

COMMENT 2: Figure 1, panel C, the staining is difficult to appreciate; please consider presenting a higher magnification image. The figure legend lettering appears to be incorrect, please verify.

RESPONSE 2: Endosialin expression in dysplastic nodules was weak and mostly restricted to the intralobular hepatic stellate cells. According to the reviewer's suggestion, we have included a high magnification image (400x) in the main revised figure. As additional information, two lower magnification images (200x) have been included in the supplementary material (revised Fig. 1 and expanded view Fig. 1). Furthermore, we thank the reviewer for alerting us of the wrong labeling in the legend. This has been corrected.

COMMENT 3: In Extended view Fig. 1, are the authors implying that there are heterogeneous expression pattern/level on TMA? Have they correlated these differences in expression with respect to HSC content in TMA sections?

RESPONSE 3: Endosialin expression in slides of whole tumor section shows considerable variability with most pronounced expression in the periphery of tumors (e.g., as shown in Fig. 1G and 1H). The small punches of TMAs are really not suitable to appreciate this variability. We therefore limited the TMA analysis to positive/negative analysis. Moreover, TMA punches are mostly taken from the tumor center and less frequently from the tumor periphery. It may therefore not be surprising that overall Endosialin expression appeared to be less prominent in the TMAs

compared to the whole tumor slides. We have previously extensively studied Endosialin expression in non-neoplastic, fibrotic and cirrhotic liver (Mogler et al, EMBO Mol Med, 2015). Here too, we saw that Endosialin is diffusely expressed by activated hepatic stellate cells.

COMMENT 4: Figure 2C: Are these differences statistically different? For the data presented in Figure 2, the number of animals used in each group is not listed in the figure legend.

RESPONSE 4: This comment was also made by reviewer 2 (comment 4): The data presented in Fig. 2C showed the percentage of mice presenting with CT-detectable tumor nodules. The data (as indicated) are indeed not statistically significant (using Fisher's test: $p=0.084$ for 6 weeks and $p=0.096$ for 7 weeks). This graph was supposed to support the hypothesis that loss of Endosialin plays a role in early tumor development. Yet, we agree with the reviewer that the simple tumor / no tumor analysis is too crude for a meaningful quantitation. We have therefore quantitated the number of CT-detectable nodules $>0.5\text{ml}$, which yields meaningful readouts and demonstrates the highly significant effect of Endosialin absence or presence on tumor growth (Fig. 2C of revised manuscript). Additionally, we have included the number of animals for each group and time point in the figure legend ($n=10$ knockout and $n=12$ wildtype animals per group and time point).

COMMENT 5: What is the meaning of figure 2P-Q? Is it supportive of the data presented in M-O?

RESPONSE 5: Figure 2P-Q of the original manuscript showed Western blot analyses of the proliferation marker PCNA in Endosialin knockout and wildtype tumor-bearing livers. It is indeed supportive of the immunohistochemical stainings shown in Fig. 2M-O. The data are normalized to total protein (scanning and evaluation was automatically done using the Amersham Western Blotting System GE Healthcare [details in methods section]) and show the quantitation of PCNA protein of each sample shown on the left. As this graphical arrangement seems to be somewhat confusing (see also comment 3 of reviewer 1), we rearranged the graphical presentation and transferred the Western blot data to the expanded view section (expanded view Fig. 7).

COMMENT 6: Figure 3, please indicate the percent downregulation of endosialin in the knockdown strategy?

RESPONSE 6: The knockdown of Endosialin was consistently more than 90% (see graph on right from 3 different transfection rounds with 3 biological replicates each; shown in S.E.M.). This information is included in the text of the methods section

COMMENT 7: Figure 3E. Is there a difference in % EdU+HCC tumor cells between non-treated (just HCC tumor cells) vs. treated with CM ns EN? Please note the poor quality of the image in panel 1 renders it difficult to appreciate the results.

RESPONSE 7: The images in panel 1 have been replaced (see also comment 4 by reviewer 1). As for the effect of conditioned medium from non-Endosialin-silenced HSC on HCC proliferation vs. HCC proliferation without conditioned medium (strictly using the same medium conditions [2%FCS/low glucose DMEM medium]), the reviewer is raising an interesting point: Non-Endosialin-silenced conditioned HSC medium indeed exerts a moderate inhibitory effect on HCC proliferation. This is an effect that is independent of Endosialin and we opted to not include this in the current manuscript, because it is focus of ongoing experiments.

2nd Editorial Decision

13 March 2017

Thank you for the submission of your revised manuscript to EMBO Molecular Medicine.

We have now received the enclosed reports from the reviewer who was asked to re-assess it. As you will see s/he is now supportive and I am pleased to inform you that we will be able to accept your manuscript pending the following final editorial amendments:

1) As per our Author Guidelines, the description of all reported data that includes statistical testing

must state the name of the statistical test used to generate error bars and P values, the number (n) of independent experiments underlying each data point (not replicate measures of one sample), and the ACTUAL p VALUE for each test (not merely 'significant' or 'P < 0.05').

2) Please provide 5 keywords in the title page.

Please submit your revised manuscript within two weeks. I look forward to receiving a revised form of your manuscript as soon as possible so that we can proceed with formal acceptance

***** Reviewer's comments *****

Referee #1 (Remarks):

All issues were adequately addressed in the revision.

2nd Revision - authors' response

13 March 2017

Thank you very much for notifying us that our revised manuscript entitled “*Hepatic stellate cells limit hepatocellular carcinoma progression through the orphan receptor endosialin*” is acceptable for publication in *EMBO Molecular Medicine* pending minor additional revisions (additional information on details of statistical analysis [exact p values, names of employed statistical tests, number of ‘n’ in each experiment]; keywords on first page of manuscript). Attached are the completed manuscript file, the updated figures, the updated extended view figures and the updated appendix.

Corresponding Author Name: Prof. Dr. Hellmut G. Augustin

Journal Submitted to: Embo Mol Med

Manuscript Number: EMM-2016-07222